# Prediction of Arrival Flight Operation Strategies under Convective Weather Based on Trajectory Clustering

**Shijin Wang \*, Jiewen Chu, Jiahao Li and Rongrong Duan**

College of Civil Aviation, Nanjing University of Aeronautics and Astronautics, No. 29 General Avenue, Nanjing 211106, China; chujiewen@nuaa.edu.cn (J.C.); lijiahao@nuaa.edu.cn (J.L.); duanrongrong@nuaa.edu.cn (R.D.)
\* Correspondence: shijin_wang@nuaa.edu.cn

**Abstract:** An airport's terminal area is the bottleneck of the air transport system. Convective weather can seriously affect the normal flight status of arrival and departure flights. At present, pilots take different flight operation strategies to avoid convective weather based on onboard radar, visual information, adverse weather experience, etc. This paper studies trajectory clustering based on the OPTICS algorithm to obtain the arrival of typical flight routes in the terminal area. Based on weather information of the planned typical flight route and flight plan information, Random Forest (RF), K-nearest Neighbor KNN (KNN), and Support Vector Machines (SVM) algorithms were used for training and establishing the Arrival Flight Operation Strategy Prediction Model (AFOSPM). In this paper, case studies of historical arrival flights in the Guangzhou (ZGGG) and Wuhan (ZHHH) terminal area were carried out. The results show that trajectory clustering results based on the OPTICS algorithm can more accurately reflect the regular flight routes of arrival flights in a terminal area. Compared to KNN and SVM, the prediction accuracy of AFOSPM based on RF is better, reaching more than 88%. On this basis, six features—including 90% VIL, weather coverage, weather duration, planned route, max VIL, and planned Arrival Gate (AF)—were used as the input features for AFOSPM, which can effectively predict various arrival flight operation strategies. For the most frequently used arrival flight operation strategies under convective weather conditions—radar guidance, AF changing, and diversion strategy—the prediction accuracy of the ZGGG and ZHHH terminal areas can exceed 95%, 85%, and 80%, respectively.

**Keywords:** terminal area; convective weather; trajectory clustering; typical flight routes; flight operation strategy

## 1. Introduction

With the increasing scale of global air traffic, abnormal flight status is becoming more prevalent, and convective weather is the primary cause of abnormal situations [1]. In 2017, 77.8% of flight delays in the US and 41.9% in the EU were related to convective weather [2]. As shown in Figure 1, the abnormal rates caused by convective weather in China from 2017 to 2021 were 51.28%, 47.46%, 46.49%, 57.31%, and 59.56%, respectively [3]. Convective weather not only causes numerous flight delays but also threatens flight safety. In general, flight operation under convective weather has always been a hot topic in civil aviation studies.

The terminal area refers to the airspace that connects the en-route space and the airport, and is responsible for the landing of arrival flights and the taking off of departure flights. As a bottleneck in the air transportation system, it has a complex airspace structure as well as multiple flying limitations, which is challenging for flight operation. When the terminal area is affected by convective weather, arrival flights choose alternative operation strategies, such as holding and diverting airborne aircraft to avoid convective weather. When the situation turns critical, a significant number of flights occupy the departure

airspace, causing ground holding or possibly ground blockage for departure flights. As a result, convective weather leads to a decrease in terminal airspace utilization, which complicates air traffic and increases the workload of pilots and controllers. The cascading effect of flight delays caused by convective weather in the terminal area can be transmitted to subsequent arrival and departure flights, reducing the operation efficiency of the terminal area and the entire air traffic system. Since arrival flight operation is more complex than departure flight operation, this paper focuses on the study of the arrival flight operation strategy in a terminal area.

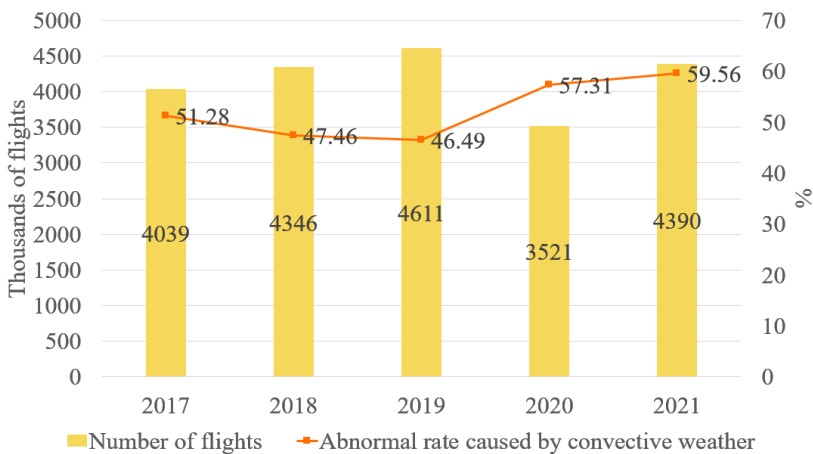

**Figure 1.** Number of flights and abnormal rate caused by convective weather from 2016 to 2020.

Although the Aeronautical Information Publication (AIP) has published a Standard Instrument Arrival (STAR) and Standard Instrument Departure (SID) for each terminal area, due to the airspace environment, traffic flow characteristics, flight habits, and other factors, the actual flight paths of a large percentage of flights are inconsistent with the procedures in practice. The clustering of historical trajectories can not only identify the prevailing traffic flow in the terminal area, but also serve as the foundation for trajectory prediction, flight path planning, and flight operation strategy prediction, which is critical for ensuring flight safety, reducing flight delay time, and improving operation efficiency of the terminal area.

Many scholars have conducted extensive research on trajectory clustering, mainly through partition algorithms, density-based clustering algorithms, and hybrid clustering algorithms. Among partition algorithms, the K-Means clustering algorithm [4–6] and the K-Medoids clustering algorithm [7,8] are mainly used. Eskstein [4] et al. applied the K-means algorithm to cluster flight trajectories and identify the main traffic flows in the terminal area, and observed the consistency of the clustering results with existing procedures to identify abnormal flight status. Xu et al. [7] used the K-Medoids algorithm for clustering, which effectively solved the mismatching problem in trajectory point selection caused by different aircraft speeds. As for density-based clustering algorithms, the DBSCAN clustering algorithm [9,10], the HDBSCAN clustering algorithm [11,12], and the RDBSCAN clustering algorithm [13] are mainly used. Murca et al. [9] used the DBSCAN algorithm to cluster flight trajectories and obtained flight paths in the terminal area. Samantha et al. [12] used the HDBSCAN clustering algorithm based on the weighted Euclidian distance function to improve the identification of air traffic flow in the terminal area. As for the hybrid clustering algorithms, Gariel et al. [14] proposed a trajectory clustering method based on principal component analysis (PCA) composed of resampling and augmented trajectory using the K-Means and DBSCAN clustering algorithm. Wang et al. [15] combined the LOF algorithm, the K-Means clustering algorithm based on time window segmentation, and the hierarchical clustering algorithm to perform trajectory clustering based on trajectory point features. Among the above algorithms, the partition algorithm was sensitive to noise and outliers, and the input parameter *k* (the number of cluster categories). Clustering results

through multiple operations may not be the same, which means the clustering results lack consistency. The density-based clustering algorithm was sensitive to two parameters: $\varepsilon$ (neighborhood radius) and MinPts (the minimum number of samples within neighborhood radius $\varepsilon$). At the same time, the density-based clustering algorithm had poor performance on the dataset with different data densities; thus, trajectory clustering performance needs to be improved in the selection of the algorithm.

At present, pilots take different flight operation strategies under convective weather conditions based on onboard radar, visual information, adverse weather experience, etc. There are two issues with pilot preference in the flight operation strategy: Different pilots have different levels of experience and risk tolerance, and different flight operation strategies exist for the same convective weather conditions. It is hard to quantify personal experience to assess the effectiveness of decision-making. Previous subjective experience cannot provide objective decision support for future flight operation strategy selection under convective weather.

Relevant studies have studied flight operation under convective weather conditions through different methods and perspectives, mainly including weather indexes research, flight deviation prediction, flight delay prediction, capacity evaluation, and flight route optimization. For the study of weather indexes, scholars have used indexes such as Combined Reflectivity (CR), Echo Top (ET), VIL [16–20], or WITI [21], and flight characteristics [22,23] to represent the influence of convective weather on flight operation. Based on these weather indexes, on the one hand, the MIT Lincoln Laboratory team used flight deviation probability [24,25] and Convective Weather Avoidance Polygon (CWAP) [26] to predict whether flights in the terminal area or en-route deviate under convective weather. On the other hand, scholars used E-WITI [27], T-WITI [28,29], and WITI/WITI-FA [30] to analyze the impacts of various convective weather conditions on air traffic delay and measure the flight delay time under convective weather. As for capacity evaluation, scholars quantified the adverse impact of convective weather on flight operation and evaluated the static or dynamic capacity of terminal area, sector, or en-route under convective weather based on the max-flow min-cut theorem [31,32], the scanline method for calculating airspace availability [33,34], and machine learning algorithm [35]. In terms of flight route optimization, previous studies mainly used path optimization algorithms such as k-shortest path algorithm [36], A* algorithm [37], as well as intelligent algorithms such as ant colony optimization algorithm [38], genetic algorithm [39], and simulated annealing algorithm [40]. Although the path planning algorithm can obtain an optimal solution, it has the disadvantages of huge calculation and being long time-consuming. Intelligent algorithms can deal with more constraints, are better in multi-objective optimization with higher solving efficiency, and widely used in flight path optimization; however, these algorithms can be easily trapped in local optimal solutions without obtaining an optimal global solution. To overcome the shortcomings of the above two kinds of algorithms, along with the rapid development of machine learning algorithms, in recent years, scholars have used machine learning algorithms [41] and hybrid algorithms [42] to conduct flight path optimization.

At present, few studies have explored the prediction of flight operation strategies under convective weather conditions. Although the studies [16–23] predicted whether flights would deviate under convective weather conditions, they did not attempt to further explore the prediction of specific flight operation strategies, providing limited support to the automatic decision of flight operation strategy. Because of the above deficiencies, this paper uses corresponding weather information from a planned typical flight route and flight plan as the features, trains on the dataset of historical arrival flights in Guangzhou (ZGGG) and Wuhan (ZHHH) terminal area through Random Forest (RF), K-nearest Neighbor (KNN), and the Support Vector Machines (SVM) algorithm to construct the Arrival Control Strategy Prediction Model (AFOSPM), and forecasts six flight operation strategies commonly used by historical flights in the terminal area. Prediction accuracy and confusion matrix are used to evaluate the prediction performance of AFOSPM, and the prediction of flight operation strategies for arrival flights through AFOSPM is verified.

The paper is organized as follows: Section 2 introduces the methods where we describe the classification of convective weather, trajectory clustering algorithm, AFOSPM, and machine learning algorithm. Section 3 describes the typical arrival flight routes of the ZGGG and ZHHH terminal area obtained by the OPTICS algorithm, which is compared to the K-Means and DBSCAN algorithm. Section 4 determines the machine learning algorithm and input features of AFOSPM, and analyses the prediction results of arrival flight operation strategies in the ZGGG and ZHHH terminal area. Section 5 offers conclusions and suggestions for future research.

## 2. Methods

### 2.1. Classification of Convective Weather

Convective weather seriously affects the take-off and landing of flights in a terminal area. The National Weather Service (NWS) classifies convective weather according to its severity, as shown in Table 1 [43]. VIL is a graphic product of the total distribution of liquid water in a vertical column with a certain bottom area in the precipitation cloud body, indicating the intensity of convective weather. ET is the height of 18.3 dBZ echo that can be detected by radar, indicating the height of convective weather. Combined with the regulations of the Civil Aviation Administration of China and airlines on flights deviating from convective weather, the weather condition below class NWS class 3 in Table 1 is taken as clear weather, and it is believed that this condition has little influence on flight operation [31]. This paper takes ET and VIL as weather features in the prediction of arrival flight operation strategies.

**Table 1.** NWS level and numerical range of weather indexes.

| NWS Level | VIL (kg/m$^2$) | ET (ft) | Type |
|:---:|:---:|:---:|:---:|
| 0 | <0.14 | | None |
| 1 | 0.14–0.7 | <25,000 | Light mist |
| 2 | 0.7–3.5 | | Moderate |
| 3 | 3.5–6.9 | | Heavy |
| 4 | 6.9–12 | 25,000–35,000 | Very heavy |
| 5 | 12–32 | | Intense |
| 6 | ≥32 | ≥35,000 | Extreme |

### 2.2. Trajectory Clustering Based on the OPTICS Algorithm

By comparing the historical trajectories with flight procedures, it was observed that even in clear weather, the radar trajectories of a considerable number of flights had an obvious distance from the arrival procedure published in AIP. Therefore, typical arrival flight routes of the terminal area were obtained by trajectory clustering to reflect flight habits. Thus, typical arrival flight routes of the terminal area were used to replace the STAR. Weather information on typical arrival flight routes were acquired as the features of AFOSPM since weather information on typical arrival flight routes has a better correlation with the flight operation strategies of historical flights. Figure 2 shows the comparison between historical trajectories and flight procedures of ZGGG and ZHHH terminal.

Trajectory clustering is based on the selection of distance calculation methods to measure trajectory similarity. Distance calculation methods mainly include the Euclidean distance, LCSS distance, DTW distance, Hausdorff distance, Fréchet distance, etc. Compared to other methods, the Fréchet distance has multiple advantages. It does not need to input calculation parameters. For trajectories of different lengths, the Fréchet distance can measure trajectory similarity without standardization and can fully consider the position and sequence relationship of trajectory points. Therefore, the Fréchet distance was selected to calculate the distance between trajectories in this paper.

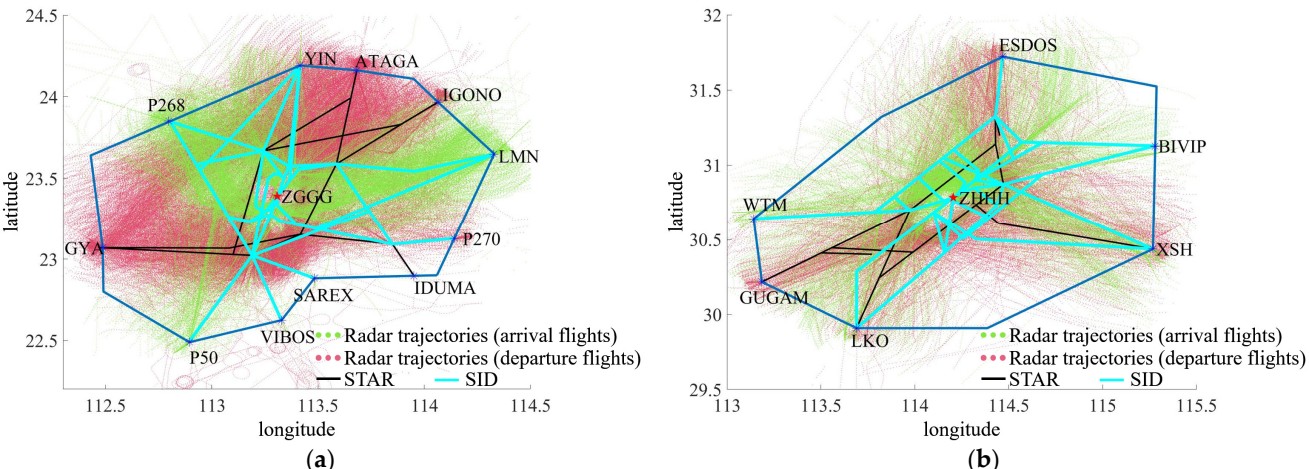

**Figure 2.** Comparison between radar trajectories and flight procedures. (**a**) ZGGG, (**b**) ZHHH.

Calculation of the Fréchet distance is as follows: For the trajectory $P$ composed of $p$ trajectory points and the trajectory $Q$ composed of $q$ trajectory points, $\sigma(P)$ and $\sigma(Q)$ represent the sequential set of trajectory points of the two trajectories respectively, then there are $\sigma(P) = (u_1, u_2, \cdots, u_p)$ and $\sigma(Q) = (v_1, v_2, \cdots, v_q)$. Thus, trajectory point pair $SC_j$ can be obtained as:

$$SC_j = \{(u_{a_1}, v_{b_1}), (u_{a_2}, v_{b_2}), \cdots, (u_{a_q}, v_{b_q})\}, \ j = 1, 2, \cdots, p \tag{1}$$

where $a_1 = j$, $b_1 = 1$, for any $i = 1, \ldots, q-1$, $a_{i+1} = a_i$ and $b_{i+1} = b_i + 1$. The distance of trajectory point pair $\left\|SC_j\right\|$ is then defined as the maximum Euclidean distance of each trajectory point pair in $p$ trajectory point pairs, as shown in Equation (2):

$$\left\|SC_j\right\| = \max_{i=1,2,\cdots,q}(dist(u_{a_i}, v_{b_i})), \ j = 1, 2, \cdots, p \tag{2}$$

Then, the Fréchet distance $D_F(P, Q)$ is defined as follows:

$$D_F(P, Q) = \min_{j=1,2,\cdots,p} \left\|SC_j\right\| \tag{3}$$

Most of the arrival trajectories in the terminal area are relatively concentrated, but there are still abnormal flight trajectories caused by various reasons. The OPTICS (Ordering Point To Identify the Cluster Structure) algorithm can obtain typical arrival flight routes in the terminal area while avoiding the influence of abnormal flight trajectories. OPTICS also has the advantages of being insensitive to input parameters and not having to specify the number of clusters in advance. The OPTICS algorithm is divided into two stages—cluster sequence generation and clustering label acquisition—as shown in Figure 3.

The process of cluster sequence generation is shown in Figure 3a. Through the input parameters of neighborhood radius $\varepsilon$ and the minimum number of samples within $\varepsilon$-radius *MinPts*, the core distance and reachable distance of each sample are calculated to generate the expansion sequence of all samples. Core distance is the minimum neighborhood radius that makes sample $x$ the core object, as shown in Equation (4). It is worth noting that this paper uses the Fréchet distance instead of the Euclidean distance when calculating the core distance. Reachable distance is the minimum neighborhood radius that makes sample $x$ the core object and sample $y$ directly reachable from sample $x$, that is, the bigger one of the core distance of sample $x$ and the Fréchet distance between sample $x$ and sample $y$, as

shown in Equation (5). In Equations (4) and (5), $N(x)$ is the number of samples that sample $x$ contains in $\varepsilon$-radius, and $D_F(y, x)$ is the Fréchet distance between sample $x$ and sample $y$.

$$coreDist(x) = \begin{cases} UNDEFINED, \ N(x) <= MinPts \\ d(x, N_\varepsilon^{MinPts}(x)), \ N(x) > MinPts \end{cases} \quad (4)$$

$$reachDist(y, x) = \begin{cases} UNDEFINED, \ N(x) <= MinPts \\ max(coreDist(x), D_F(y, x)), \ N(x) > MinPts \end{cases} \quad (5)$$

The process of clustering label acquisition is shown in Figure 3b. By inputting the clustering threshold $\varepsilon_i$, the reachable distance and core distance of each sample are compared with $\varepsilon_i$, and the clustering label of each sample is generated to finish the trajectory clustering.

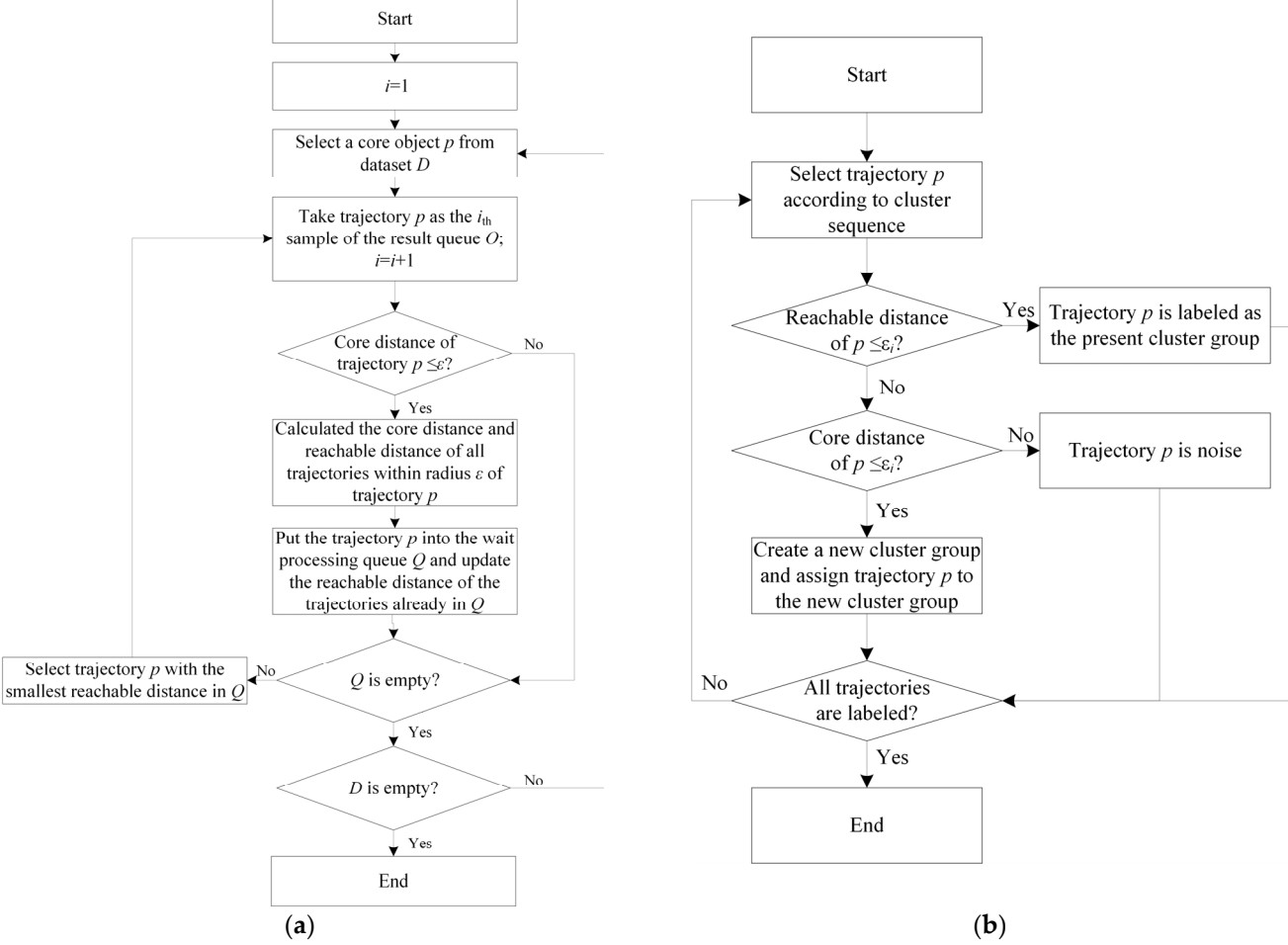

**Figure 3.** Trajectory clustering process based on the OPTICS algorithm. (**a**) Cluster sequence generation, (**b**) Clustering label acquisition.

### 2.3. Categories of Arrival Flight Operation Strategies

The arrival segment refers to the flight section from AF to the Initial Approach Fix (IAF). Arrival flight operation strategy refers to the flight operation strategy taken by the pilot from AF to IAF point. The arrival flight operation strategies are mainly of seven types: no-impact, radar guidance, airborne holding, AF changing, IAF point changing, intrusion, and diversion. Figure 4 shows examples of seven types of arrival flight operation strategies.

No-impact strategy means that when the level of convective weather on the flight plan route is NWS class 2 or below, the flight flies according to the flight plan route, as shown in Figure 4 ①. Radar guidance strategy means that the pilot selects the airspace

with good weather conditions to complete the arrival process through radar guidance. The actual flight route under the radar guidance strategy is relatively random, as shown in Figure 4 ②. The airborne holding strategy means that the flight waits in the air. After the weather conditions improve, it still flies according to the flight plan route, as shown in Figure 4 ③. This strategy needs to judge the trend and moving speed of convective weather according to the weather forecast, which can be implemented when the holding time is not long. The AF changing strategy is similar to the IAF point changing strategy. The AF changing strategy means that the flight enters the terminal area from another AF instead of flight plan AF, as shown in Figure 4 ④. The IAF point changing strategy means that the flight flies to another IAF point following a different flight route instead of the flight plan route, as shown in Figure 4 ⑤. Intrusion strategy means that the flight crosses the convective weather area and still flies according to the flight plan route, as shown in Figure 4 ⑥. The diversion strategy means that after the flight bypasses the convective weather area on the flight plan route, it returns to the flight plan route and continues to fly, as shown in Figure 4 ⑦.

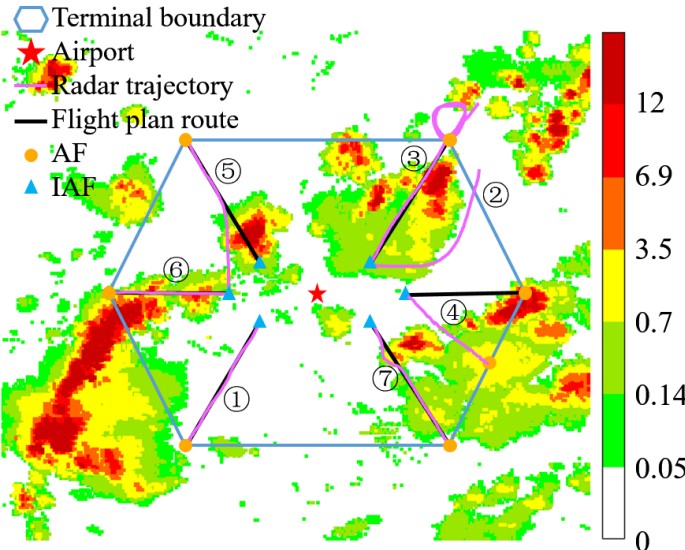

**Figure 4.** Explanation of arrival flight operation strategies.

In the actual flight process, some flights may adopt two or more flight operation strategies during the arrival segment, such as a combination of intrusion and diversion, AF changing, and no-impact. According to the intensity, coverage, and duration of convective weather, and considering the workload of controllers brought by flight time with different arrival flight operation strategies, this paper ranks seven flight operation strategies through the impacts of flight operation strategies on the workload of controllers: radar guidance > airborne holding > AF changing > IAF point changing > diversion > intrusion > no-impact. Each historical arrival flight is defined to only one flight operation strategy.

### 2.4. AFOSPM and Machine Learning Algorithms

Figure 5 shows the AFOSPM construction process. The planned typical flight route is determined according to flight plan AF, airport operation direction, and arrival or departure airport. The weather information on the planned typical flight route and flight plan information regarded as features together with the label are input into the machine learning algorithm to obtain the preliminary AFOSPM, and the output is the predicted flight operation strategy. By calculating the error between the predicted flight operation strategy and the real flight operation strategy, the parameters of the machine learning algorithm are updated and the AFOSPM is retrained. The training is stopped after reaching the set training times to obtain the final AFOSPM.

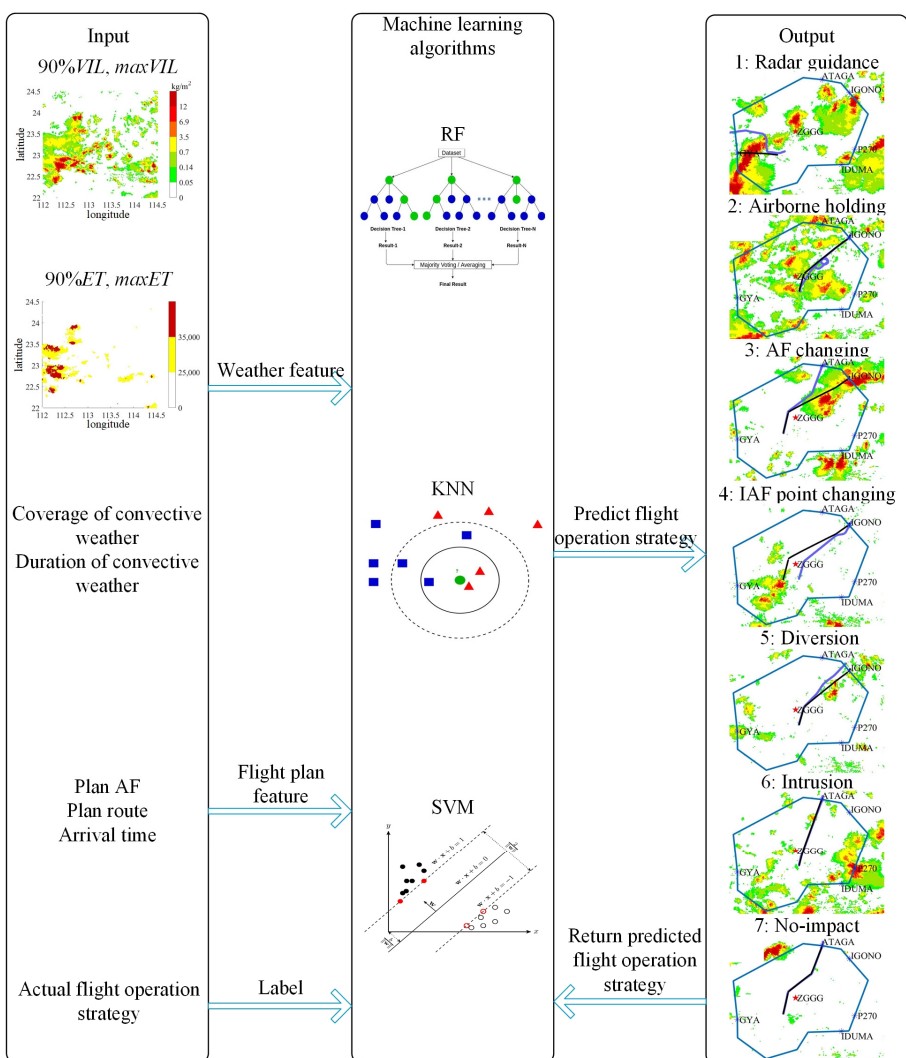

**Figure 5.** AFOSPM construction process.

Machine learning algorithms have the powerful ability to learn from numerous historical flight trajectories and match the internal relationship between convective weather and flight operation strategy. The AFOSPM obtained by machine learning algorithm can deal with complex nonlinear characteristics and realize the prediction of arrival flight operation strategies under convective weather conditions. In this paper, three machine algorithms, RF, KNN, and SVM, were used to obtain AFOSPM.

The RF algorithm is based on the Decision Tree (DT) algorithm in the bagging framework. Figure 5 shows the training and testing process of the RF algorithm. It is assumed that the number of training samples is $N$ and the number of features is $M$; then sample $N$ times from $N$ training samples in the way of putting back samples to form a training subset for training a single DT. $m$ features are randomly selected from $M$ features, among which $m$ should be far less than $M$. The optimal splitting of DT is calculated according to these $m$ features. When acquiring DT, each node is split this way until it is no longer able to split, that is, the training of a single DT is completed. The whole DT is formed without a pruning operation, and finally, the training of $n$ sub-models obtained by the DT algorithm is completed. When testing, the test samples are input into each sub-model, and $n$ predicted values given by each sub-model are obtained. The final predicted values of the test samples are obtained by majority voting of these $n$ predicted values in the classification problem.

KNN directly compares the samples of the testing set and the training set. Its core idea is to calculate the distance between each sample of the testing set and all the samples of the training set, and then sort the distances to select $K$ points with the smallest distance, which

is to select $K$—most similar samples of the training set in the feature space. Finally, through the labels of the selected training samples, the predicted value of each test sample can be obtained according to the principle of majority voting. As shown in Figure 5, the green dot is a sample of the testing set, while the blue square and red triangle represent the two types of data in the training set. When $K = 3$, the predicted value of the sample of the testing set is the red triangle, and when $K = 5$, the predicted value of the sample of the testing set is the blue square.

The SVM algorithm aimed to obtain the separation hyperplane with the largest geometric interval that can correctly partition the training dataset. As shown in Figure 5, the black and white circles represent two kinds of data. The red solid circle and the red hollow circle are the closest samples for two kinds of data, which are support vectors for building a separation hyperplane. $w \cdot x + b = 0$ is the separation hyperplane that maximizes the geometric interval between two types of data, where $w$ is the normal vector of the hyperplane, determining the direction of the hyperplane, $b$ is the offset of the hyperplane, determining the position of the hyperplane. During the test process, the feature $x$ of the test sample is plugged in $w \cdot x + b$, and the predicted value is obtained according to the result value greater than 1 or less than $-1$.

### 3. Trajectory Clustering Results

This paper selects flight data from 3 August to 31 August 2018, including flight plan, historical radar trajectory, and ET and VIL weather data. Flight plan data includes flight ID, registration number, aircraft type, departure airport, destination airport, flight plan path, and other information, which is used for data filtering before trajectory clustering, planned typical flight route determination, and input of AFOSPM. Historical radar trajectory data updates every 8 s, including time, flight ID, height, longitude, latitude, and other information, which is used for trajectory clustering and the determination of arrival flight operation strategy for historical flights. ET and VIL data updates every 6 min, and it is stored in the form of $0.01° \times 0.01°$ (longitude $\times$ latitude), which is used for the determination of arrival flight operation strategy for historical flights and the calculation of weather information as input of AFOSPM.

The ZGGG and ZHHH terminal area are selected as the research subjects in this paper. Guangzhou Baiyun Airport is the third busiest airport in China. Its terminal area is located in the busy traffic zone of the Pearl River Delta, and its airspace structure is complex with a lot of arrival routes. Located in the subtropical zone, the temperature is high in August and convective weather occurs frequently. Wuhan Tianhe Airport is the second busiest airport in China. Its terminal area is the main transportation hub in central China, connecting the north–south and east–west air traffic. It also has a lot of arrival routes and a complex airspace structure. It is located in the subtropical zone, with high temperatures and frequent convective weather in August. Figure 6 shows the airspace structure of the ZGGG and ZHHH terminal area.

In this paper, 10,551 arrival flights of ZGGG and 6345 arrival flights of ZHHH from 3 August to 31 August 2018, are selected for trajectory clustering. Figures 7 and 8 show the clustering results of the ZGGG and ZHHH terminal area based on the OPTICS algorithm. The blue polygon in the figure is the boundary of the terminal area, the blue asterisk point is AF, and the red five-pointed star is the location of the airport. Different colors represent different typical arrival flight routes. Figure 9 shows the number and ratio of flights under convective weather and clear weather in typical arrival flight routes, and the number and ratio of flights that are not clustered into any typical arrival flight route. Colors used in Figure 9 correspond to Figures 7 and 8, representing flights under clear weather, and grey represents flights under convective weather or not clustered into any typical arrival flight route. As can be seen from Figures 7–9, although the number of trajectories and directions of typical arrival flight routes for two terminal areas are different, the OPTICS algorithm identifies typical arrival flight routes while eliminating outliers.

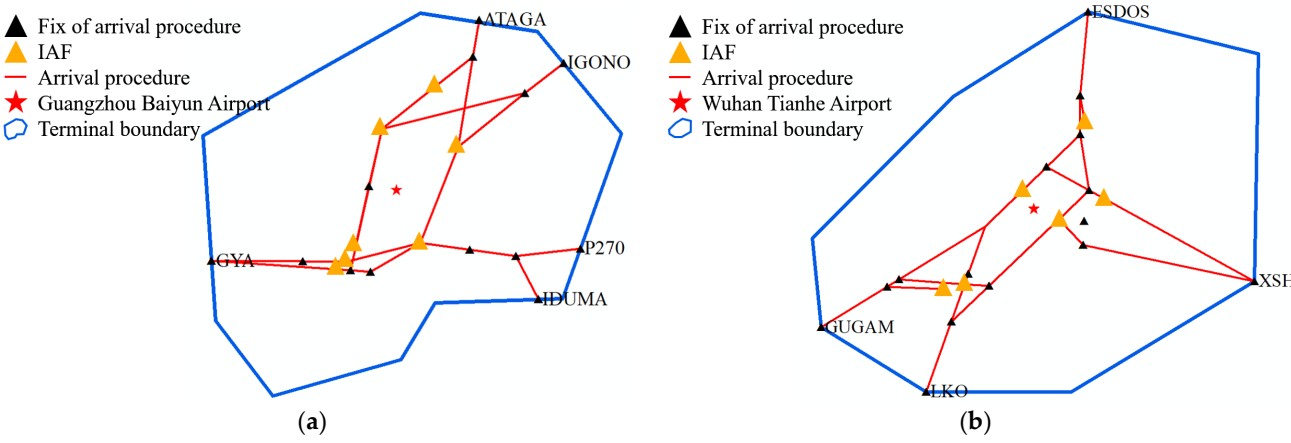

**Figure 6.** Airspace structure. (**a**) ZGGG, (**b**) ZHHH.

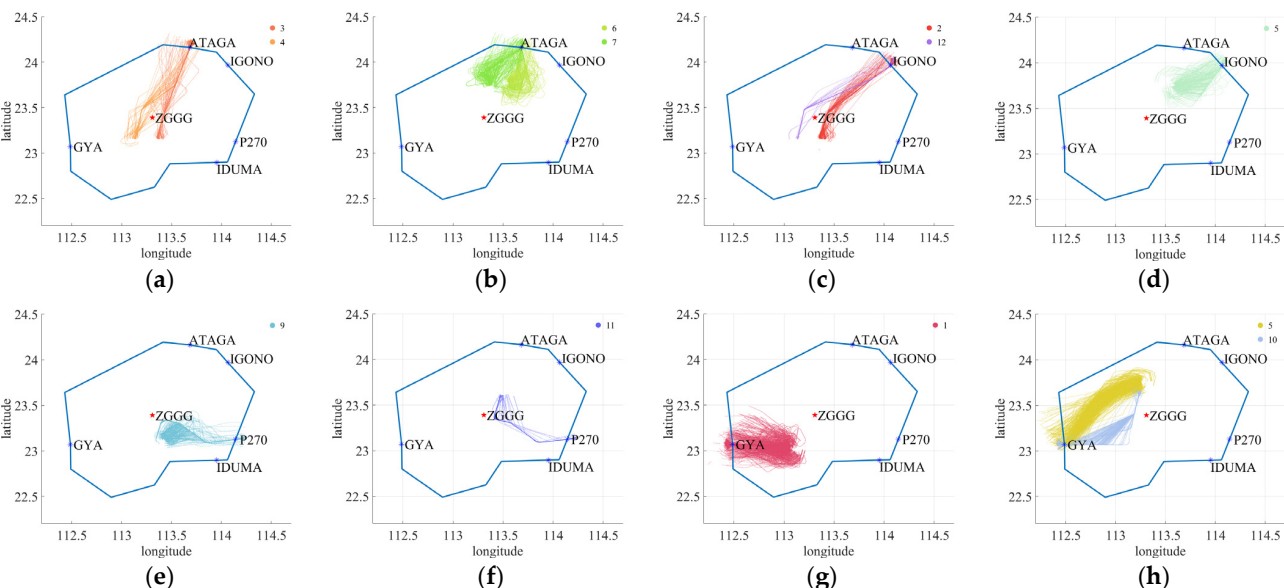

**Figure 7.** Clustering results of flight trajectories of ZGGG (OPTICS). (**a**) AG: ATAGA, Direction: North; (**b**) AG: ATAGA, Direction: South; (**c**) AG: IGONO, Direction: North; (**d**) AG: IGONO, Direction: South; (**e**) AG: P270, Direction: North; (**f**) AG: P270, Direction: South; (**g**) AG: GYA, Direction: North; (**h**) AG: GYA, Direction: South.

To verify the clustering results of the OPTICS algorithm, K-Means and DBSCAN algorithms are used to cluster the same dataset. According to the clustering results of the OPTICS algorithm, the *k* value of the K-Means algorithm is determined. Figures 10–13 show the clustering results of the ZGGG and ZHHH terminal area obtained by the K-Means algorithm and the DBSCAN algorithm, respectively. As can be seen from Figures 11–14, although K-Means and DBSCAN algorithms can identify typical flight routes in different directions, there remain two problems. First, K-Means and DBSCAN algorithms merge some typical flight routes into one typical flight route. For example, for the ZGGG terminal area, No. 5 and No. 10 typical flight routes in Figure 7h can be distinguished by OPTICS. In Figures 10 and 12, these two typical flight routes are merged into one typical flight route. For the ZHHH terminal area, No. 1, No. 5, and No. 9 typical flight routes in Figure 9e,f can be distinguished by OPTICS. In Figures 11 and 13, these three typical flight routes are merged into one typical flight route. The inaccurate clustering results of the K-Means and DBSCAN algorithm affect the calculation of weather information on the planned typical flight route, which can further reduce the prediction accuracy of flight operation strategies. Secondly, K-Means and DBSCAN algorithms do not separate some typical flight

routes of different arrival directions. For example, No. 2 and No. 7 typical flight routes in Figure 9g,h are two typical flight routes that originated at XSH with different arrival directions. In Figure 11, the K-Means algorithm merges these two typical flight routes into three typical flight routes without distinguishing the arrival directions. In Figure 13, the DBSCAN algorithm also merges these two typical flight routes into three typical flight routes, regardless of the arrival direction. In conclusion, the typical flight routes obtained by the OPTICS algorithm are more accurate and effective.

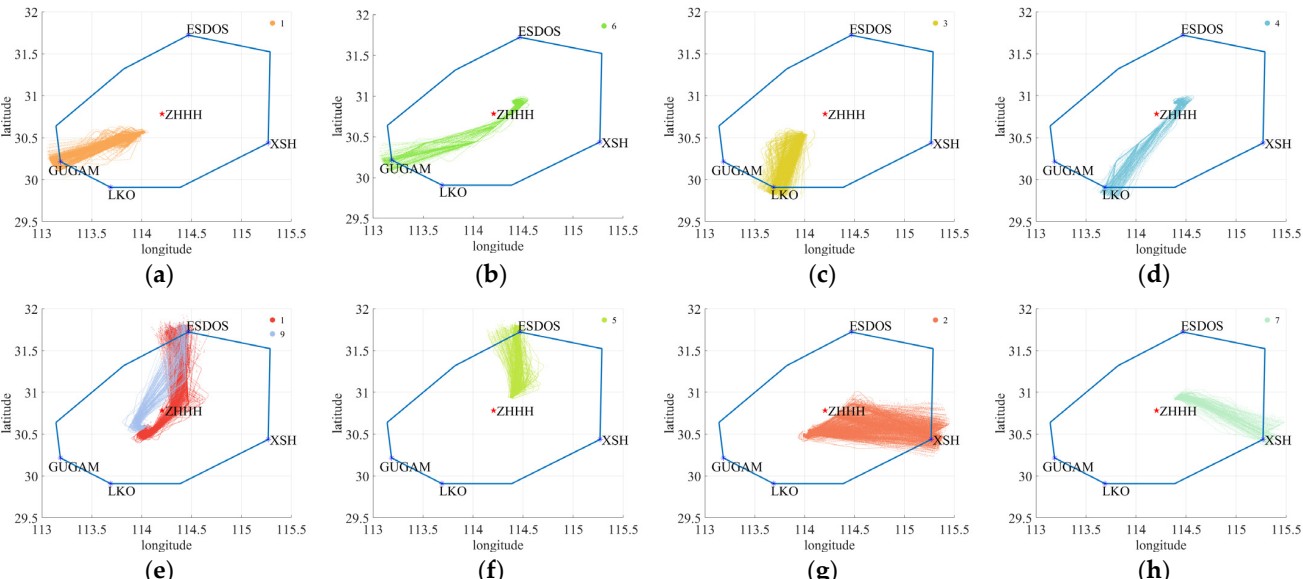

**Figure 8.** Clustering results of flight trajectories of ZHHH (OPTICS). (**a**) AG: GUGAM, Direction: North; (**b**) AG: GUGAM, Direction: South; (**c**) AG: LKO, Direction: North; (**d**) AG: LKO, Direction: South; (**e**) AG: ESDOS, Direction: North; (**f**) AG: ESDOS, Direction: South; (**g**) AG: XSH, Direction: North; (**h**) AG: XSH, Direction: South.

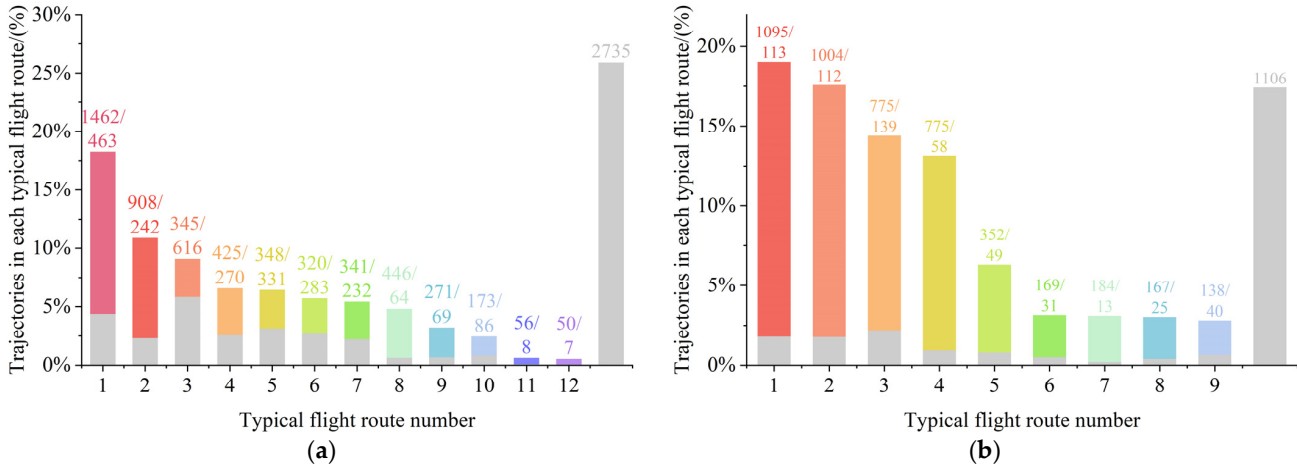

**Figure 9.** Percentage of flight trajectories on typical flight routes. (**a**) ZGGG; (**b**) ZHHH.

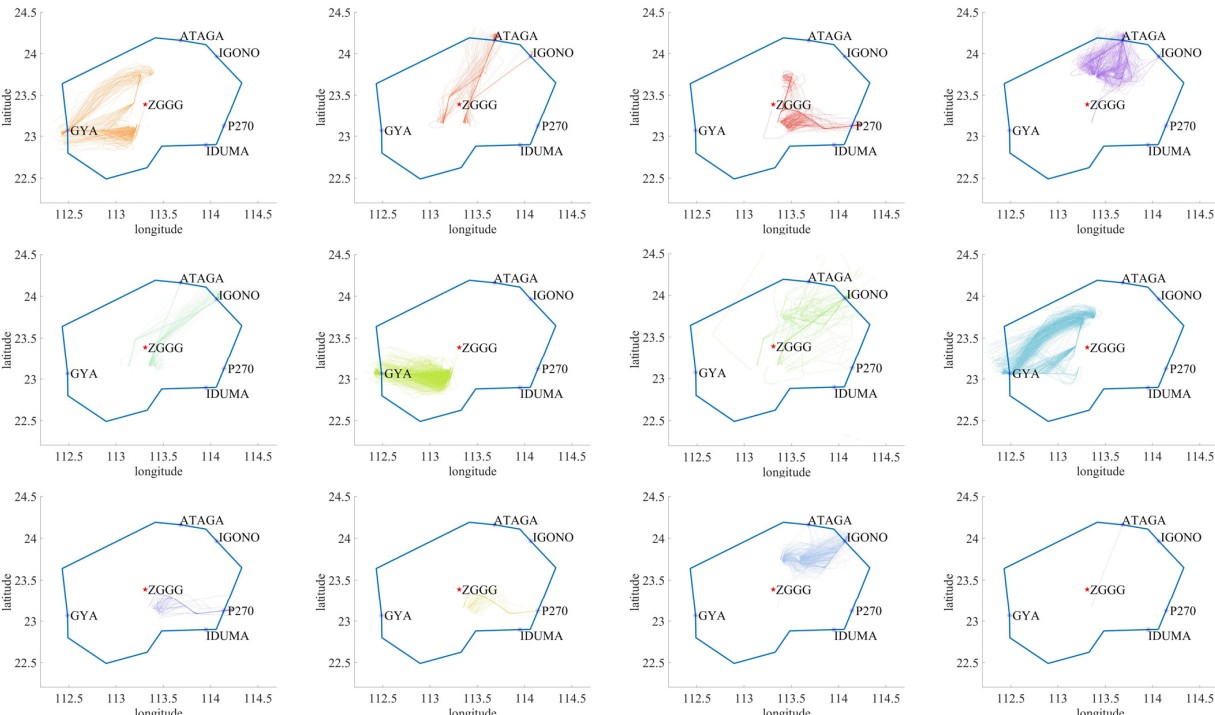

**Figure 10.** Clustering results of flight trajectories of ZGGG (K-Means).

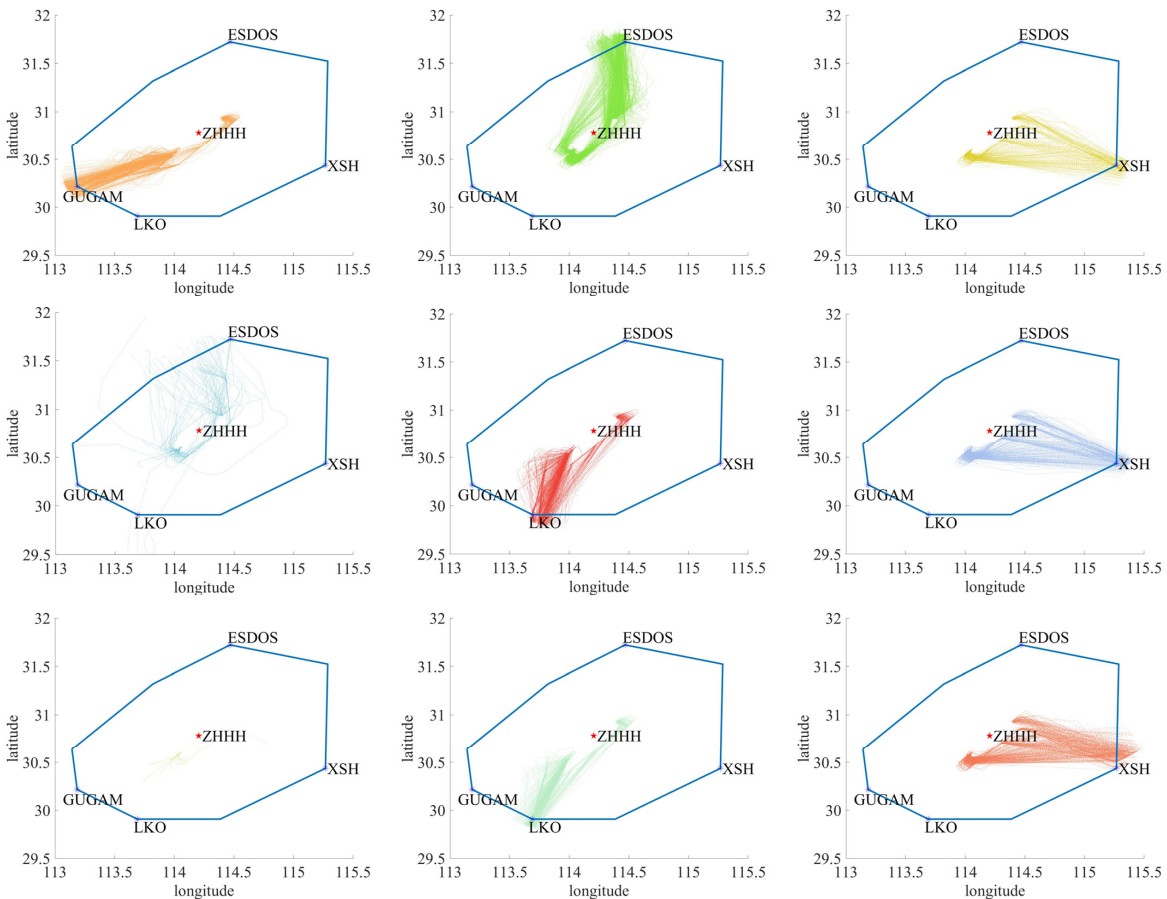

**Figure 11.** Clustering results of flight trajectories of ZHHH (K-Means).

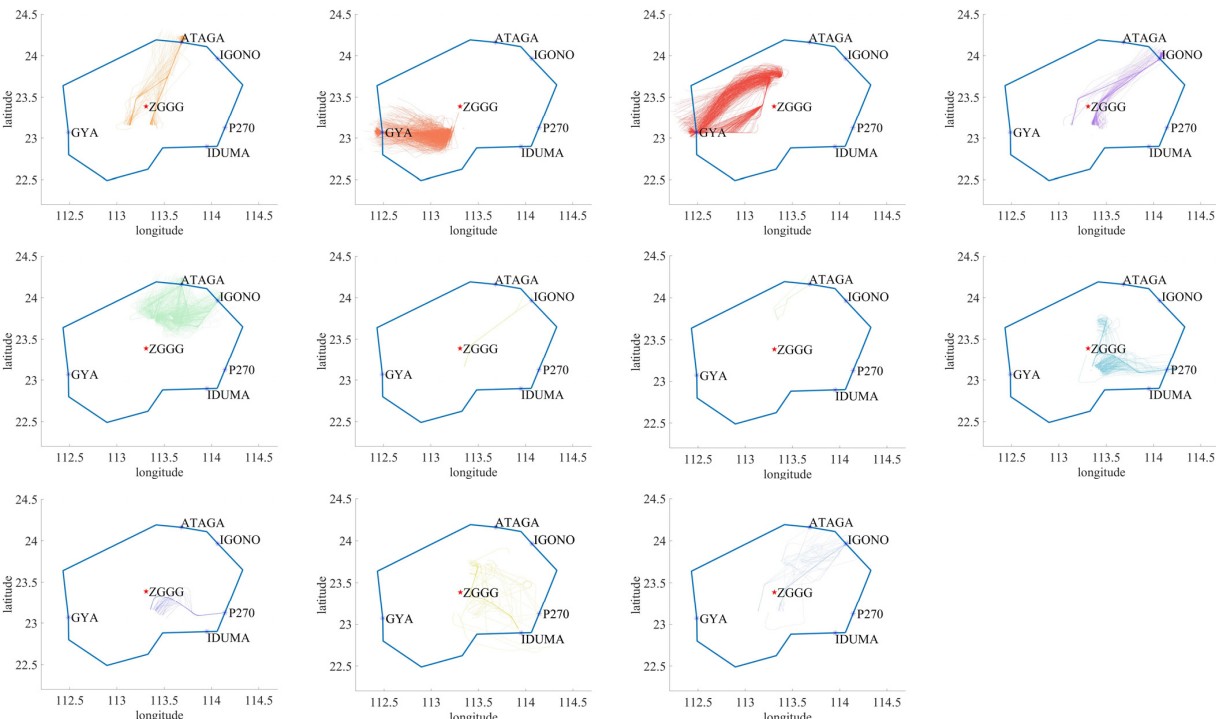

**Figure 12.** Clustering results of flight trajectories of ZGGG (DBSCAN).

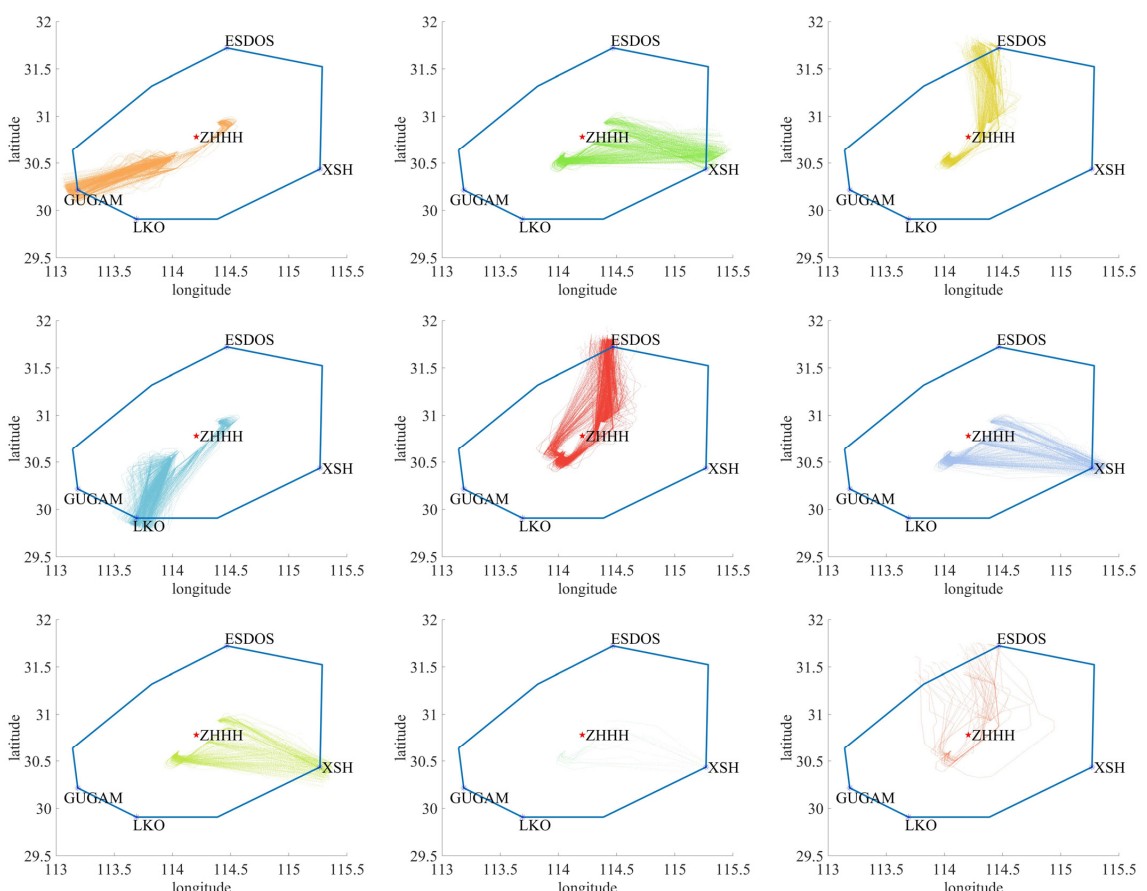

**Figure 13.** Clustering results of flight trajectories of ZHHH (DBSCAN).

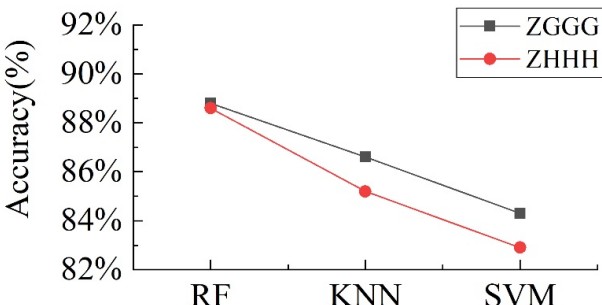

**Figure 14.** Comparison of the prediction accuracy of the three algorithms.

## 4. Prediction Results of Arrival Flight Operation Strategies

After obtaining the typical flight routes clustered by the OPTICS algorithm, the flight operation strategy and the planned typical flight route of each flight can be acquired through radar trajectory, the typical flight route that each flight belongs to, the flight plan, and the weather condition when each flight arrives. Afterward, the planned typical fight route can be calculated according to the flight operation strategy each flight takes, arrival direction, and the weather condition when each flight arrives. The real arrival flight operation strategy and the planned typical fight route of each flight is the basis for prediction, which makes further research possible.

In this section, 5405 and 1685 arrival flights under convective weather situations in the ZGGG and ZHHH terminal area are predicted, respectively, and AFOSPM with higher prediction accuracy is constructed from two aspects, which are algorithm comparison and feature selection, and the similarities and differences of prediction results between the ZGGG and ZHHH terminal area are analyzed.

In terms of algorithm comparison, RF, KNN, and SVM algorithms in machine learning are used to build AFOSPM. The 90th percentile value of VIL within the planned typical flight route, the proportion of the value of VIL in NWS class 3 and above within the planned typical flight route, the duration that the proportion of the value of VIL in NWS class 3 and above within the planned typical flight route is 10% and above, the planned typical flight route, the maximum percentile value of VIL within the planned typical flight route, and flight plan AF are taken as the input features, and the actual arrival flight operation strategy of each flight is taken as the label. The optimal parameters of each algorithm are determined by maximizing the prediction accuracy of AFOSPM. The learner type of RF algorithm is DT, and the number of the learner is 30. The $K$ value of the KNN algorithm is 3, and the input features are standardized before prediction. The kernel function of the SVM algorithm is Gaussian kernel, the training time is 500, and the input features are standardized before prediction. Figure 14 shows the prediction accuracy of the three machine learning algorithms using 10-fold cross-validation. Compared with the other two algorithms, the RF algorithm shows better performance on datasets of the ZGGG and ZHHH terminal area, so the RF algorithm is selected to establish AFOSPM.

In terms of feature selection, as shown in Table 2, nine features representing weather and flight plan information are selected. To reach the highest prediction accuracy, only one feature is added in each experiment based on the last experiment, and the change in prediction accuracy after adding this feature is shown to represent the influence of this feature on the prediction of arrival flight operation strategies. Figures 15 and 16 show the change in prediction accuracy while adding features using 10-fold cross-validation. Prediction accuracy increased with the addition of features, and prediction accuracy reached 88.8% for the ZGGG terminal and 88.6% for the ZHHH terminal while using 90%VIL, Coverage, Duration, Plan route, Max VIL, and Plan AF as input features. The prediction performance is limited and even reduced by adding more features. Therefore, the first six features in Table 2 are selected for the establishment of AFOSPM.

**Table 2.** Features and meaning.

| Feature | Meaning |
|---------|---------|
| 90%VIL | The 90th percentile value of VIL within the planned typical flight route |
| Coverage | The proportion of the area with the value of VIL in NWS class 3 and above within the planned typical flight route |
| Duration | The duration that the proportion of the area with the value of VIL in NWS class 3 and above within the planned typical flight route is 10% and above |
| Plan route | The planned typical flight route |
| Max VIL | The maximum percentile value of VIL within the planned typical flight route |
| Plan AF | The flight plan AF |
| Arrival time | The planned time when the flight arrives at the terminal area boundary |
| 90%ET | The 90th percentile value of ET within the planned typical flight route |
| Max ET | The maximum percentile value of ET within the planned typical flight route |

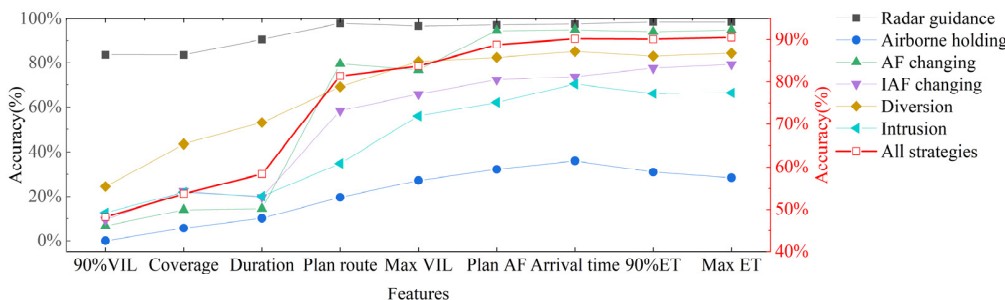

**Figure 15.** Prediction accuracy change caused by feature selection (ZGGG).

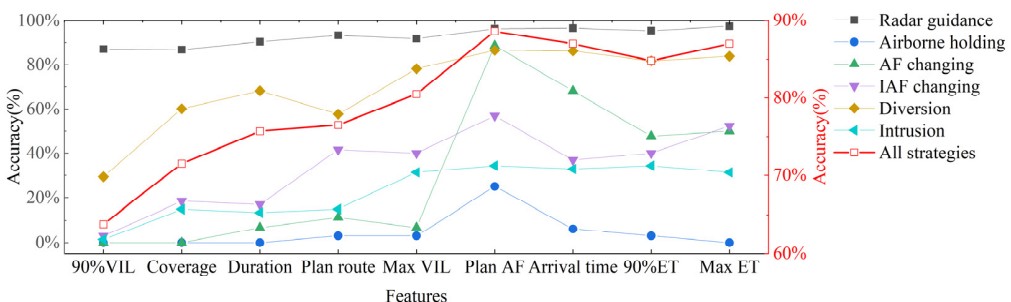

**Figure 16.** Prediction accuracy change caused by feature selection (ZHHH).

Figure 17 shows the confusion matrix of prediction results of arrival flight operation strategies in the ZGGG and ZHHH terminal area, where 1–6 correspond to radar guidance, airborne holding, AF changing, IAF point changing, diversion, and intrusion, respectively. In the datasets of the ZGGG and ZHHH terminal area, the sum proportion of radar guidance, AF changing, and diversion strategy is 83.4% and 90.2%, respectively. The prediction accuracy of radar guidance strategy, AF changing, and diversion strategy is more than 95%, 85%, and 80%, respectively. Meanwhile, the prediction accuracy of airborne holding and intrusion is below 80%. This is because when convective weather conditions in NWS class 3 and above exist on the planned typical flight route, more flights select radar guidance, AF changing, and diversion strategy rather than airborne holding and intrusion strategy; thus, AFOSPM has difficulty identifying part of flights adopting airborne holding and intrusion strategy through the input features. In addition, the prediction accuracy of the IAF point changing strategy is relatively low, because the proportion of IAF point changing strategy is very low in all kinds of convective weather conditions, and the number of flights adopting the IAF point changing strategy is relatively small, so the prediction accuracy of the IAF point changing strategy is not as good as radar guidance, AF changing, and diversion strategy.

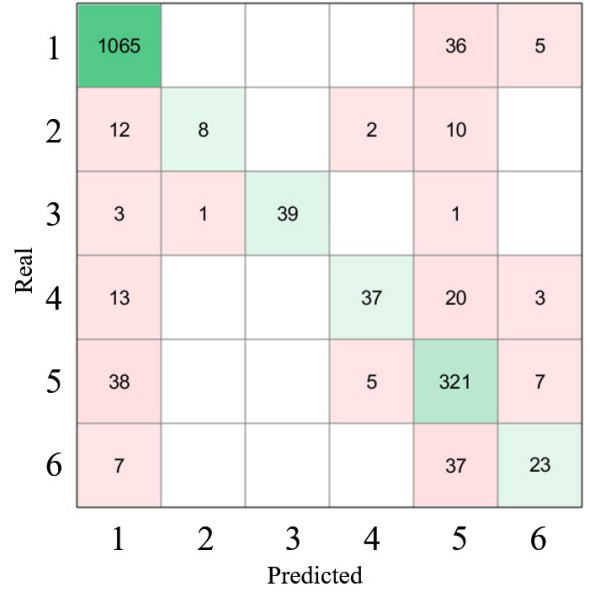
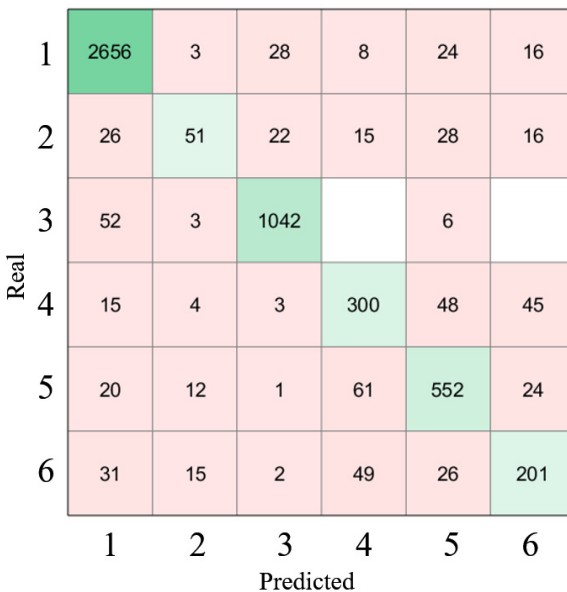

**Figure 17.** Confusion matrix of prediction results of arrival flight operation strategies. (**a**) ZGGG; (**b**) ZHHH.

In addition, by comparing the prediction results of the ZGGG and ZHHH terminal area, it was found that only the prediction accuracy of diversion strategy of the ZGGG terminal area is 4% lower than that of the ZHHH terminal area, while the prediction accuracy of other strategies is higher than that of the ZHHH terminal area, among which the intrusion strategy is 28% higher, the IAF point changing strategy is 15% higher, the airborne holding strategy is 7% higher, the AF changing strategy is 6% higher, and radar guidance strategy is 1% higher. This may be because there are more samples in the dataset of the ZGGG terminal area, and the arrival flight operation strategies adopted by flights in the ZGGG terminal area are more correlated with convective weather. Therefore, in the prediction of arrival flight operation strategies, it is better to select a terminal area with more significant convective weather and more data samples as the research object.

## 5. Conclusions

In this paper, typical flight routes are obtained by clustering historical flight trajectories. Arrival flight operation strategies are predicted based on the weather information on planned typical flight routes and flight plan information to provide decision support for the flight operation strategy in future convective weather conditions in a terminal area. Classification of convective weather, trajectory clustering based on the OPTICS algorithm, categories of arrival flight operation strategies, and AFOSPM and machine learning algorithms are first described. Then, the ZGGG and ZHHH terminal areas are taken as research objects to obtain typical arrival flight routes based on the OPTICS algorithm. Finally, AFOSPM is established using the RF algorithm. Conclusions can be drawn as follows:

(1) Compared with the K-Means and the DBSCAN algorithm, the OPTICS algorithm can distinguish typical flight routes with close distance and different airport operation directions.

(2) In terms of algorithm selection, compared with KNN and SVM, the prediction accuracy of AFOSPM based on RF is better, reaching more than 88%. In feature selection, 90%VIL, Coverage, Duration, Plan route, Max VIL, and Plan AF are selected as the input features of AFOSPM among nine features representing weather and flight plan information. The prediction accuracy of AFOSPM on the dataset of the ZGGG and ZHHH terminal area is 88.6% and 88.8%, respectively.

(3) With the input of 90%VIL, Coverage, Duration, Plan route, Max VIL, and Plan AF, for radar guidance, AF changing and diversion strategies with a combined frequency of more than 80% under convective weather conditions, the prediction accuracy of AFOSPM of the ZGGG and ZHHH terminal area is more than 95%, 85%, and 80%, respectively.

(4) By comparing the prediction results of arrival flight operation strategies in ZGGG and ZHHH, only the prediction accuracy of diversion strategy in the ZGGG terminal area is 4% lower than that in the ZHHH terminal area while the prediction accuracy of other strategies is 1–28% higher than that in the ZHHH terminal area. This may be because the frequency, intensity, and coverage of convective weather are higher in the ZGGG terminal area, which leads to a stronger correlation between arrival flight operation strategies and convective weather. Therefore, a terminal area with more significant convective weather has better prediction performance of flight operation strategies.

On this basis, on the one hand, future research will further study the algorithm of prediction of flight operation strategies and extend it to the prediction of departure flight operation strategies. On the other hand, according to the prediction of terminal flight operation strategies, the flight diversion path can be optimized, and the take-off and landing time can be predicted, which can be used to further calculate the expected flight time delay. It provides a new research direction for flight time optimization in a terminal area under convective weather conditions.

**Author Contributions:** Conceptualization, S.W., J.C. and J.L.; methodology, S.W. and J.L.; validation, S.W. and J.C.; formal analysis, S.W. and J.C.; investigation, S.W., J.C. and J.L.; data curation, J.C., J.L. and R.D.; writing—original draft preparation, S.W.; writing—review and editing, J.C., J.L. and R.D.; visualization, J.L.; supervision, S.W. All authors have read and agreed to the published version of the manuscript.

**Funding:** This research received no external funding.

**Institutional Review Board Statement:** Not applicable.

**Informed Consent Statement:** Not applicable.

**Data Availability Statement:** Not applicable.

**Acknowledgments:** We would like to thank the New Generation Intelligent ATC Laboratory of Nanjing University of Aeronautics and Astronautics for providing the radar trajectory data, weather data, and flight plan data used in the trajectory clustering and the prediction of arrival flight operation strategies.

**Conflicts of Interest:** The authors declare no conflict of interest.

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
