# Peer review of "Prediction of Arrival Flight Operation Strategies under Convective Weather Based on Trajectory Clustering"

_aerospace, doi:10.3390/aerospace9040189_

Round 1
Reviewer 1 Report
Some general remarks
- The subject is of interest for the aviation community and should be persued.
- The mathematical methodology is sound and results are not questionable.
- The presentation of what the authors wanted to communicate is veiled by a) poor use of English grammar, b) some missing explanations, c) too few and too poor graphics, d) some open questions probably emerging from a)-c) but solely in the aviation part of the paper.
- There might be a general misunderstanding of the decision making process in aviation, which must be clarified. In general, pilots follow the agreed flight path unless they are forced by convection to deviate. The intention to deviate from the flight plan is based a) on the onboard radar and the radar reflectivity values of the clouds ahead, b) visual information in the out-the-window scene, c) the pilots adverse weather experience, d) aircraft limitations, e) risk preception, f) and also some stress factors. The pilots request for a heading change, and/or however more rare an altitude change, is approved or rejected by ATC in a possibly longer dialogue with a final solution. Thus the observed scatter in flight trajectories under convective weather is dominantly caused by the pilot and to a lesser degree by ATC in the joint decision finding process. The authors identify the source of trajectory scatter with the controller and denote the trajecory scatter as a result from different control strategies. The reviewer disagrees here and requests clarification.
- Figures are generally too small.
- Detailed comments
- the term "abnormal flight" does not exist in aviation, but "abnormal flight status", "abnormal situation", "abnormal operation" and similar expressions. Please explain what is intended to show and check the term throughout the paper.
- Change of flight height. Line 156. VIL does not reveal the cloud top height. Thus the cloud might still be under the actual flight level and the pilot will overfly the cloud. ET is important for the pilot and to eliminate it weakens the methodology. The reference to the TMA height is irrelevant here.
- The reviewer does not understand the statements from line 319 to 335.
- Figure 10 a) and b) need a explanatory caption.
- Figures 9,11,12 should be enlarged so that the individual trajectories , at least the outliers, can be identified as lines rather than as part of a point cloud.
- The result of trajectory clustering remains vague. If trajectory clustering should replace a STAR route (L 164), then a unique route must be calculated . This , however, is not addressed in the paper but touched only in line 353 by mentioning the "route centerline". This topic needs clarification as well.
- There are many English grammar errors, too many to list them all.
- l. 23-25 what is the subject what the predicate?
- what does " the accuracy of G. and W. terminal area" mean?
- l 116 should read "being time-consuming"
- l 419-420 conditions .... exists, (must read "exist")
- l 80 must read "the hierarchical..."
Author Response
Dear reviewer:
Thank you for your comments on our manuscript entitled “Prediction of arrival control strategies under convective weather based on trajectory clustering”. We fully read all comments, and those comments are very valuable and helpful for revising and improving our paper. We have made corrections to every comment, which we hope to meet with your approval. Revision is expressed using the “Track Changes” function, and the ‘line’ in the following statement refers to the newly uploaded manuscript. The responses to your comments are as follows:
1. Thank you for your appreciation of our subject. We’ll keep on working on the corresponding research in the future.
2. Thank you for your recognition of the mathematical methodology and results. Your approval is significant and valuable for the efforts we made.
3. a) We feel apologize for the poor grammar in our manuscript. We’ve read the manuscript completely and carefully for the last few days to correct the grammar mistakes. For example, in line 188, ‘trajectory’ is changed into ‘trajectories’. In line 209, ‘is’ is changed into ‘are’. In line 459, ‘shows’ is changed into ‘show’. In line 468 and line 470, ‘Accuracy change’ is changed into ‘Prediction accuracy changing’. In line 496, ‘is’ is changed into ‘are’.
b) We added missing explanations at certain positions. In line 187 to 196, we explained why we need to cluster the typical flight routes and the reason we used the typical flight routes to replace flight procedures. In line 338 to 348, we added the application of flight plan data, radar trajectory data, and weather data. In line 360 to 382, we added the explanations of Figure 7-9.
c) We increased some graphics, such as Figure 2, Figure 7, and Figure 8. We enriched the expression in some graphics. For example, in Figure 4, we described the features in detail, and added several graphics under the demonstration of features, algorithms and flight operation strategies. In Figure 10 and Figure 11, we split the clustering results obtained by K-Means and DBSCAN algorithms into separate trajectories for clearer observation of each individual.
d) We read through the whole manuscript, corrected all the mistakes that we found, and added some expressions at certain positions. For example, we updated the data in Figure 1 at line 45. We added the description of the terminal area in line 47 to 48. We revised the explanation of the problems existing in the pilot’s preference in the flight operation strategy at present in line 103 to 113. We corrected the usage of ET in line 183 to 184.
4. After reading this comment, we have a more profound understanding of the decision making process in aviation according to your abundant comprehension and vivid description. We do have a misunderstanding on it. Thus, we ensure the opinion that the pilots are mostly responsible for deviating from the flight plan, and the strategy within the terminal should be divided and explained from the view of pilots instead of controllers. As a revision, we corrected ‘control strategy’ into ‘flight operation strategy’. The ACSPM is changed into AFOSPM. We read the whole manuscript carefully and changed all the corresponding descriptions to ensure the statement is accurate.
5. We enlarged Figure 1-6, Figure 9, and Figure 17 to ensure that the above figures are clear enough to meet your suggestion.
6. a) We changed all the terms ‘abnormal flight’ in line 36, line 37, line 39, and line 78 into ‘’ abnormal flight status.
b) We deeply apologize for our carelessness. ET reveals the height information of convective weather, which is an essential weather index. As the height of the terminal area is from the ground surface to 6000m normally, it’s very important to add ET into the prediction of flight operation strategies. We do experiment with the effect of ET in the feature selection, which is from line 454 to 470. But we carelessly forget to express the usage of ET in line 181 to 182. Thus, we corrected our statement in line 183-184, where we expressed that VIL and ET are both the weather features that we use in this manuscript.
c) Line 319-335 in the old version of the manuscript gives unnecessary details and lengthy expressions. We simplify the expression in line 360 to 382 in the new version of the manuscript, only describing the number of trajectories that we use, the content in Figure 7-9, and the performance of the OPTICS algorithm in the new version of the manuscript.
d) We added the explanation of Figure 10 in the old version of the manuscript (Figure 9 in the new version of the manuscript) in line 376 to 378.
e) In the old version of the manuscript, we put the clustering results into a single graphic, in which some typical flight routes are hard to recognize. So we draw the typical flight routes obtained by the OPTICS algorithm for each arrival gate and each arrival direction. There are 12 typical flight routes for the ZGGG terminal and there are 9 typical flight routes for the ZHHH terminal. For the clustering results obtained by the K-Means and DBSCAN algorithm, we draw them separately, which is clearer for readers to identify the problems of the K-Means and DBSCAN algorithm.
f) We expressed the importance of typical flight routes in line 187 to 196, and we explained how to calculate the real arrival flight operation strategy and the planned typical flight route in line 421 to 428. After acquiring the planned typical flight route of each flight, the weather information on the planned typical flight route can be calculated. In our manuscript, the purpose that we use arrival typical flight routes to replace STAR is that the weather information on the planned typical flight route is regarded as features of AFOSPM, which is described in line 192-196.
g) (1) We changed ‘the accuracy’ into ‘the prediction accuracy’. The subject is ’The prediction accuracy’ and the predicate is ‘can’.
(2) This grammar mistake is corrected in line 135.
(3) This grammar mistake is corrected in line 479.
(4) This grammar mistake is corrected in line 93.
All in all, we tried our best to improve the manuscript and made some changes according to your suggestion. We appreciate your kind and practical comments earnestly, and we hope the correction will meet with approval.
Thank you for your comments and suggestions sincerely!

Reviewer 2 Report
The article is well written and i suggest acceptance.
Author Response
Thank you for your appreciation and approval of the manuscript!
Round 2
Reviewer 1 Report
The authors responded to the reviewers comments satisfactorily. Despite of some room for improvement the paper should be published as it is.